# Triplet correlations in Cooper pair splitters realized in a two-dimensional electron gas

Qingzhen Wang[1,6], Sebastiaan L. D. ten Haaf [1,6], Ivan Kulesh[1], Di Xiao[2], Candice Thomas[2], Michael J. Manfra [2,3,4,5] & Srijit Goswami [1] ✉

Cooper pairs occupy the ground state of superconductors and are typically composed of maximally entangled electrons with opposite spin. In order to study the spin and entanglement properties of these electrons, one must separate them spatially via a process known as Cooper pair splitting (CPS). Here we provide the first demonstration of CPS in a semiconductor two-dimensional electron gas (2DEG). By coupling two quantum dots to a superconductor-semiconductor hybrid region we achieve efficient Cooper pair splitting, and clearly distinguish it from other local and non-local processes. When the spin degeneracy of the dots is lifted, they can be operated as spin-filters to obtain information about the spin of the electrons forming the Cooper pair. Not only do we observe a near perfect splitting of Cooper pairs into opposite-spin electrons (i.e. conventional singlet pairing), but also into equal-spin electrons, thus achieving triplet correlations between the quantum dots. Importantly, the exceptionally large spin-orbit interaction in our 2DEGs results in a strong triplet component, comparable in amplitude to the singlet pairing. The demonstration of CPS in a scalable and flexible platform provides a credible route to study on-chip entanglement and topological superconductivity in the form of artificial Kitaev chains.

Coupling two normal leads to a superconductor can give rise to non-local transport processes directly involving both leads. Two opposite-spin electrons from a Cooper pair in the superconductor can be split into the leads via a process known as Cooper pair splitting (CPS). The dominant transport mechanism that gives rise to CPS is crossed Andreev reflection (CAR), whereby a higher order process allows two electrons to be injected simultaneously into the superconductor to form a Cooper pair. Additionally, a single electron can tunnel through the superconductor from one lead to the other through a process known as elastic co-tunnelling (ECT). The ability to control these processes has important implications for two distinct fields. Firstly, efficient CPS can be used to generate spatially separated entangled electrons, that can be used to perform a Bell test[1–6]. Secondly, in the context of topological superconductivity, it has been shown that CAR

and ECT are crucial ingredients required to implement a Kitaev chain[7] using quantum dot-superconductor hybrids[8,9].

CPS has been studied in various mesoscopic systems coupled to superconductors, such as semiconductor nanowires[10–13], carbon nanotubes[14,15], and graphene[16]. Quantum dots (QDs) are generally added between the leads and the superconductor. The charging energy of the QDs ensures that electrons forming a Cooper pair preferentially split into separate dots, rather than occupying levels in the same dot. This results in correlated electrical currents at the two normal leads. It has thus far been challenging to independently measure the relevant virtual processes (i.e. ECT and CAR) and isolate them from local processes, such as normal Andreev reflection or direct tunnelling via sub-gap states. In a set of recent studies on hybrid nanowires, it was shown that these challenges could be overcome to

[1]QuTech and Kavli Institute of Nanoscience, Delft University of Technology, Delft 2600 GA, The Netherlands. [2]Department of Physics and Astronomy, Purdue University, West Lafayette 47907 IN, USA. [3]Elmore School of Electrical and Computer Engineering, Purdue University, West Lafayette 47907 IN, USA. [4]School of Materials Engineering, Purdue University, West Lafayette 47907 IN, USA. [5]Microsoft Quantum Lab, West Lafayette 47907 IN, USA. [6]These authors contributed equally: Qingzhen Wang, Sebastiaan L. D. ten Haaf. ✉e-mail: s.goswami@tudelft.nl

create a highly efficient Cooper pair splitter[17] and to realize a minimal Kitaev chain[18]. A key idea is that the QDs were coupled via extended Andreev bound states (ABSs) in the semiconductor-superconductor hybrid[19–21], rather than the continuum above the superconducting gap. Therefore, by controlling the ABS energy with electrostatic gates, it was possible to tune the relative amplitudes of ECT and CAR. These developments pave the way for more advanced experiments, where the geometrical constraints of 1D systems will pose restrictions on the complexity of possible devices. An ideal platform to overcome these restrictions are semiconductor 2DEGs. Not only do they offer flexibility in device design, but also serve as a scalable platform to create and manipulate topologically protected Majorana bound states in artificial Kitaev chains.

We demonstrate here for the first time the observation of Cooper pair splitting in a 2D semiconductor platform. This is achieved by coupling two quantum dots via a hybrid proximitized section in an InSbAs 2DEG. By applying an external magnetic field, we polarize the spins of the QDs, allowing us to use them as spin-filters. This, in combination with highly efficient CPS, allows us to accurately resolve the spin of the electrons involved in CAR and ECT. The large spin-orbit coupling in our 2DEGs, in combination with the device dimensions, results in significant spin precession for the electrons. Importantly, we show that this leads to strong equal-spin CAR currents that are of similar amplitude to the conventional opposite-spin processes. Through rotation of the magnetic field angle relative to the spin-orbit field, we show that the ECT and CAR processes can be tuned to equal amplitudes, satisfying a key requirement for realizing a Kitaev chain in semiconductor-superconductor hybrids.

## Results

### Device and characterization

Devices are fabricated on an InSbAs 2DEG with epitaxial aluminum grown by molecular beam epitaxy. This material has been established to have a low effective mass, high g-factor and large spin-orbit coupling[22,23]. Figure 1a, b illustrate the device structure together with the three-terminal measurement circuit. The two depletion gates (pink) define a quasi-1D channel of about 150 nm, contacted on each

side with normal leads. The middle of the channel is proximitized via a 150 nm-wide aluminium strip (green), which is kept electrically grounded. Quantum dots on the left and right are created using the finger gates (blue) and the ABS energy is controlled by the central ABS gate (purple). The biases $V_L$ and $V_R$ applied to the left and right leads can be varied independently. The currents $I_L$ and $I_R$ in the left and right leads are measured simultaneously. We define a positive current as the flow of electron charge from the leads to the superconductor.

First, the two innermost finger gates are used to define tunneling barriers on either side of the hybrid region. Figure 1c, d show the measured local conductance $G_{RR} = \frac{dI_R}{dV_R}$ and non-local conductance $G_{LR} = \frac{dI_L}{dV_R}$ as a function of the ABS gate voltage $V_{ABS}$. The induced gap in the hybrid section is found to be $\Delta_{ind} \approx 220\,\mu eV$. The correspondence between $G_{RR}$ and $G_{LR}$ shows the presence of an extended discrete ABS in the proximitized section. The observed sign-switching in the non-local signal is typical for an extended ABS probed in a three-terminal measurement[24–26]. Next, two quantum dots are created on either side of the proximitized section. Their electro-chemical potentials are controlled by applied voltages $V_{QDL}$ and $V_{QDR}$. The charge stability diagrams of both QDs (Fig. 1e, f) show Coulomb diamonds with clear even-odd spacing. The pair of Coulomb peaks show linear splitting as a function of magnetic field, indicative of a spin-degenerate single orbital level (Fig. S1). The superconducting gap $\Delta_{ind}$ is clearly visible at the charge degeneracy points, indicative of a weak coupling to the proximitized region[27,28]. Charging energies of QDL and QDR are 1.9 meV and 1.4 meV respectively, much larger than the induced superconducting gap.

### CAR and ECT

For CAR, an electron from each of the two leads is simultaneously transferred to the superconductor via an extended ABS to form a Cooper pair (Fig. 2a). This should therefore result in positively correlated currents in the leads ($I_L = I_R$). For ECT (Fig. 2b), an electron from the left or right lead tunnels to the opposite lead via the hybrid section, which should thus give rise to negatively correlated currents ($I_L = -I_R$). As we will show below, by controlling the QD levels and voltage biases, it is possible to distinguish currents arising from ECT and CAR. Such

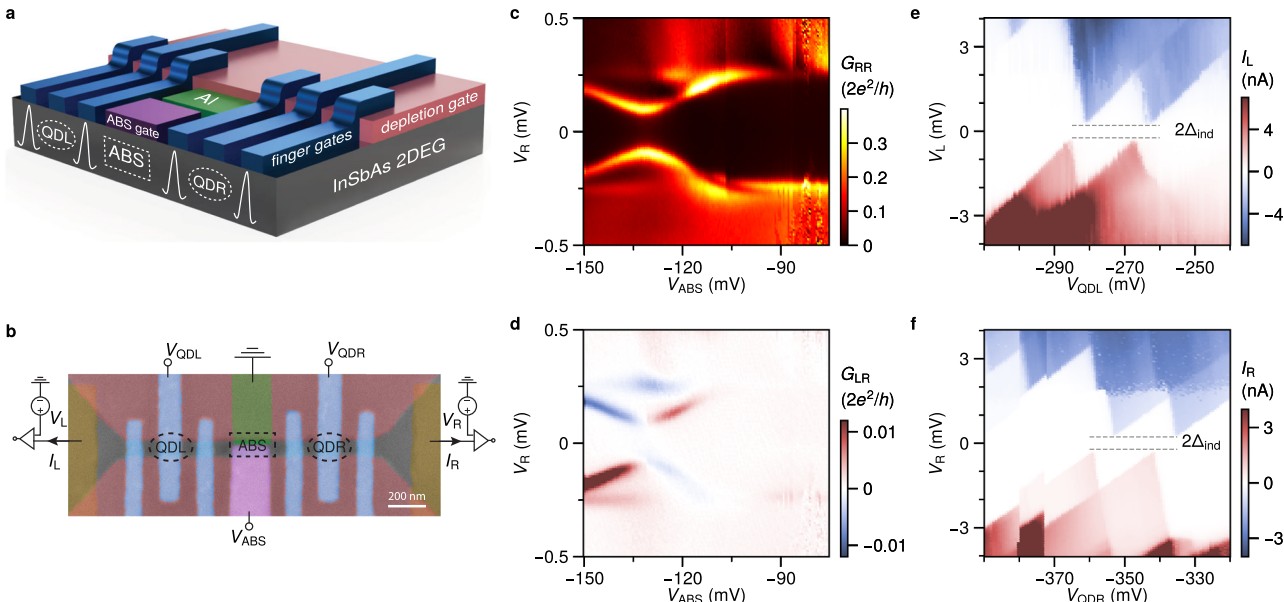

**Fig. 1 | Basic device characterization. a** A 3D illustration of the device. The two quantum dots (QDL and QDR), and the region hosting ABSs are indicated. For clarity, the gate-dielectric layers are not shown. **b** False-color scanning electron micrograph of Device 1, including a schematic of the circuit diagram for three-terminal measurements. Tunneling spectroscopy measurements showing (**c**) local conductance $G_{RR}$ and (**d**) non-local conductance $G_{LR}$ as a function of the ABS gate voltage $V_{ABS}$ and right bias voltage $V_R$. Coulomb diamonds of the QDs are measured for (**e**) QDL and (**f**) QDR.

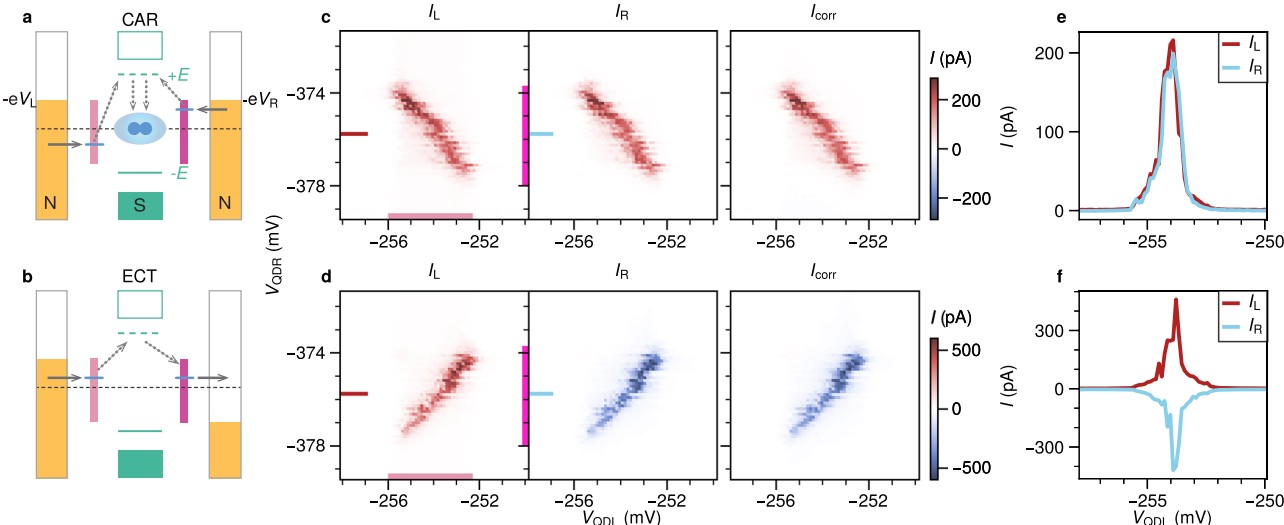

**Fig. 2 | Correlated CAR and ECT signals.** Diagrams of the transport cycles for (**a**) CAR and (**b**) ECT. Blue lines indicate the energies of the QD levels required for transport via the ABS at energy $\pm E$. Purple bars mark the energy window in which transport is allowed, corresponding to the marked regions in the measurement panels (**c** and **d**). **c** Charge stability measurement of QDL and QDR with $V_L = V_R = -120\,\mu V$ taken at $V_{ABS} = -245\,mV$. Equal currents with the same sign are observed at the left ($I_L$) and right ($I_R$) leads only when the QD energy levels are anti-aligned, as expected for CAR. **d** Repeated measurement, but with $V_L = -V_R = -120\,\mu V$. Equal currents with opposite sign are observed only when the QDs are aligned in energy, as expected for ECT. The correlated currents $I_{corr}$ are calculated from $I_L$ and $I_R$ as described in the main text. Exemplary line traces at $V_{QDR} = -375\,mV$ for CAR and ECT are plotted in (**e**) and (**f**) respectively.

measurements are shown in Fig. 2c, d. Here $V_{QDL}$ and $V_{QDR}$ are each tuned close to a selected charge degeneracy point and the currents $I_L$ and $I_R$ are simultaneously measured. The large charging energies of the dots ensure that each lead strongly prefers accepting or donating a single electron. We further ensure that the applied biases are lower in energy than any sub-gap states in the hybridized region, such that local transport is suppressed. To demonstrate CAR, we set $V_L = V_R = -120\,\mu V$ and sweep $V_{QDL}$ and $V_{QDR}$. A finite current is observed only along a line with negative slope, for both $I_L$ and $I_R$ (Fig. 2c). Furthermore, the currents are equal both in magnitude and sign (Fig. 2e). Converting the gate voltages to electro-chemical potentials ($\mu_L, \mu_R$), we confirm that CAR mediated transport occurs when $\mu_L = -\mu_R$ (Fig. S4c). This is consistent with the requirement that the energies of the electrons forming the Cooper pair must be equal and opposite. To demonstrate ECT, we apply biases with opposite polarity ($V_L = -V_R = -120\,\mu V$). Unlike CAR, a finite current is observed only along a line with positive slope (Fig. 2d). This is consistent with energy conservation during ECT, which demands that $\mu_L = \mu_R$. Furthermore, the currents are now equal in magnitude, but opposite in sign (Fig. 2f). Note that when biasing only $V_L$ or $V_R$ and grounding the other lead, both ECT and CAR become visible in the charge stability diagram (Fig. S2).

Importantly, for both CAR and ECT we observe no notable current when the bias and energy conditions are not met, indicating that unwanted local processes are strongly suppressed. In combination with strongly correlated currents, this suggests a relatively large signal-to-noise ratio of the CPS process. To characterize this, we calculate the CPS efficiency and visibility (Fig. S4). Following[15,17], we obtain a combined CPS efficiency above 90%, on par with the highest previously reported values[15,17]. Applying a larger bias that exceeds the sub-gap state energy (but is still below $\Delta_{ind}$) results in additional local, non-correlated signals which only depend on a single QD (Fig. S3) and significantly reduce the CPS efficiency.

To systematically characterize the CAR and ECT measurements, we calculate the correlated current $I_{corr} \equiv \mathrm{sgn}(I_L I_R) \cdot \sqrt{|I_L||I_R|}$ (Fig. 2c, d)[17]. It is non-zero only when $I_L$ and $I_R$ are both non-zero and thus highlights features mediated by ECT and CAR. Furthermore the sign of $I_{corr}$ clearly distinguishes CAR (always positive) from ECT (always negative).

## Zero field spin blockade

In the absence of a magnetic field, the orbital levels of the QDs are spin-degenerate. Therefore, if the dot has an even number of electrons, the first electron to occupy the next orbital (a transition denoted as $0 \leftrightarrow 1$) can be either spin-up or spin-down. However, to add the second electron ($1 \leftrightarrow 2$), the Pauli exclusion principle requires it to have an opposite spin. The effect of this spin-filling rule leads to a blockade of transport, which depends on the nature of the underlying process.

We first focus on ECT in the $(-,+)$ bias configuration, denoting that a negative bias is applied to the left lead and a positive bias is applied to the right lead (Fig. 3b). When the QDs are tuned to the $(0 \leftrightarrow 1, 1 \leftrightarrow 2)$ transition, a situation can arise where the left QD is occupied with e.g. a spin-up electron (coming from the left lead), whereas the right QD can only accept a spin-down electron (since the spin-up state has already been occupied). At this point transport from left to right is blocked, analogous to the well-known Pauli blockade in double quantum dots[29]. This spin blockade is clearly seen when the QDs are tuned over successive charge transitions. In Fig. 3c we see that the ECT current is suppressed for the $(0 \leftrightarrow 1, 1 \leftrightarrow 2)$ transition. Reversing the bias polarities to $(+,-)$, a similar blockade is observed for the $(1 \leftrightarrow 2, 0 \leftrightarrow 1)$ transition, as expected (see Fig. 3e).

In the $(-,-)$ configuration, only CAR mediated transport can occur and we find a suppression in CAR current for the $(0 \leftrightarrow 1, 0 \leftrightarrow 1)$ transition. This is a direct consequence of the Cooper pairs in an s-wave superconductor having a singlet pairing. Thus, for transport to occur, each QD must donate an electron of *opposite* spin in order to create a singlet Cooper pair in the superconductor. Transport is therefore blocked when both dots are occupied by electrons with the same spin (Fig. 3a). Finally, in the $(+,+)$ configuration a blockade is expected for the $(1 \leftrightarrow 2, 1 \leftrightarrow 2)$ transition (Fig. 3d), as observed in the measurements. Qualitatively similar measurements of spin blockade for CAR and ECT are presented for another device (Fig. S8). We note that a finite amount of current remains for each blockaded transition, indicating the presence of a spin-relaxation mechanism in our system. The hyperfine interaction is one such mechanism that can lift the Pauli blockade[30,31]. We confirm this by applying a magnetic field to suppress the spin-mixing due to the hyperfine interaction, and find that 35 mT is sufficient to fully suppress the remaining current (Fig. S5).

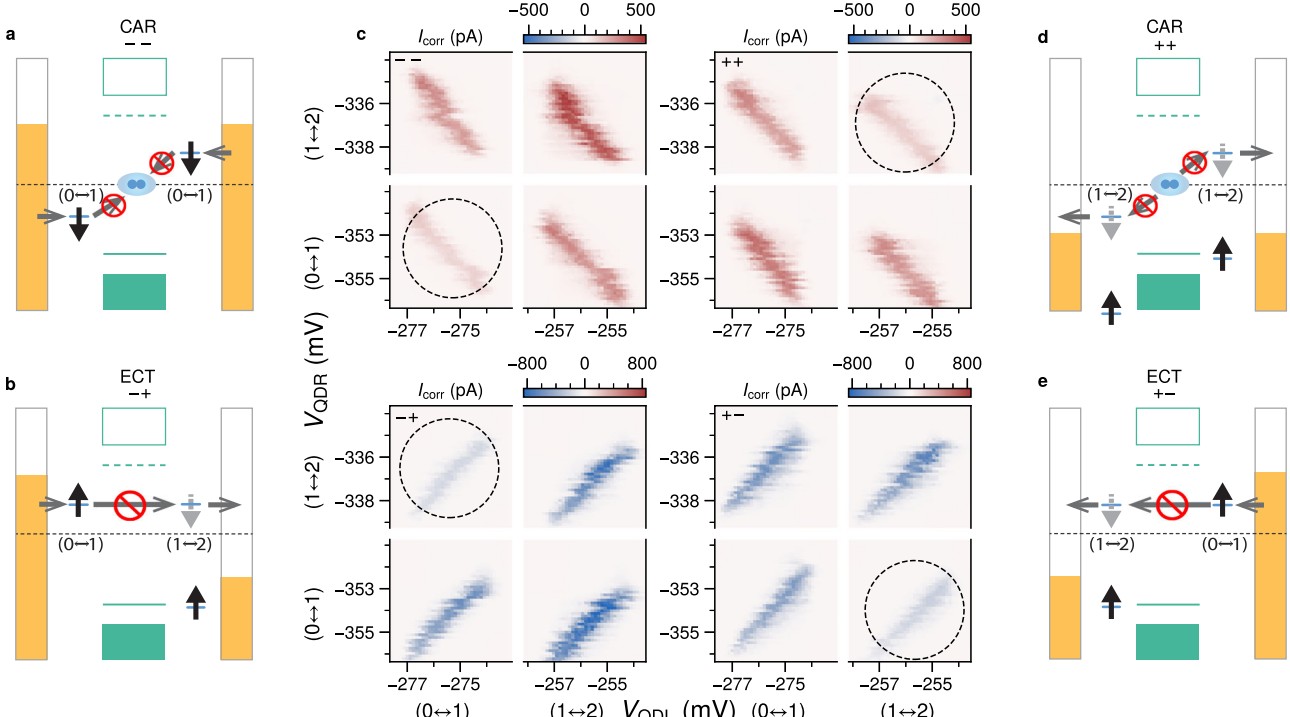

**Fig. 3 | Spin blockade at zero magnetic field.** Charge stability diagrams are obtained for all four bias polarity combinations, to measure either CAR ($V_L = V_R$) or ECT ($V_L = -V_R$). **a**, **b**, **d** and **e** show energy diagrams illustrating situations expected to lead to transport blockades. Arrows within the dots either represent an already occupied spin state (black), or a state available to be occupied by an incoming electron (grey). **c** The corresponding measurements of $I_{corr}$ plotted against $V_{QDL}$ and $V_{QDR}$, with applied biases $|V_L| = |V_R| = 120\,\mu V$ and $V_{ABS} = -220\,mV$. The used bias polarity for each set of measurements is noted in the top left corner. For each bias configuration a specific transition is suppressed (dashed circles) as a consequence of the blockade depicted alongside the measurement. Gate voltage ranges are interrupted to zoom-in on the relevant ECT and CAR features.

## Singlet and triplet ECT/CAR

The spin degeneracy of the QD levels is lifted by applying a magnetic field, allowing us to operate them as spin-filters (Fig. S1). When the Zeeman splitting exceeds $|eV_L|$, $|eV_R|$, only spin-up (↑) electrons are involved in transport at a (0 ↔ 1) transition and only spin-down (↓) at a (1 ↔ 2) transition. In the absence of spin-orbit coupling, CAR is only expected to occur when both QDs are tuned to host electrons with opposite spin. The opposite applies to ECT, where a current is only expected when the QDs are tuned to receive electrons with equal spin.

As shown in Fig. 4b, when an in-plane field of 150 mT is applied along $B_y$ (i.e. perpendicular to the channel), CAR current is only present in the quadrants where the electrons have opposite spins (↑↓ and ↓↑) and completely suppressed for the equal-spin (↑↑ and ↓↓) configuration. Similarly, no current is detected for opposite-spin ECT, while transport is allowed for equal-spin ECT. This spin-dependent transport indicates that the direction of the spin-orbit field $B_{SO}$ is along $B_y$, making spin a good quantum number. This is also consistent with the expected Rashba spin-orbit interaction in a quasi-1D channel with momentum along the z-direction and electric field perpendicular to the 2DEG plane. Applying the magnetic field perpendicular to $B_{SO}$ (i.e. along $B_z$), a spin-up electron may acquire a finite spin-down component, due to the spin-orbit interaction. The consequence of this can be seen in Fig. 4c, where we now observe sizeable currents for equal-spin CAR and opposite-spin ECT. The full evolution of the spin-specific ECT and CAR currents can be obtained by performing an in-plane rotation of the magnetic field (Fig. 4d). The averaged amplitudes of equal-spin CAR and opposite-spin ECT currents $\langle I_{corr} \rangle$ are found to oscillate smoothly between full suppression at $\theta \approx 90°$ and 270° ($\mathbf{B} \| \mathbf{B}_{SO}$), and their maximum strength at $\theta \approx 0°$ and 180° ($\mathbf{B} \perp \mathbf{B}_{SO}$). This result does not depend on a specific choice of orbitals in the QDs (Fig. S6).

The ability to accurately resolve the spin of the electrons in CPS is particularly relevant in the context of entanglement witnessing. An important metric capturing this, is the spin cross-correlation[3,4]. As described in[32], we calculate the spin cross-correlation from the measured currents as:

$$C = \frac{(I^{\uparrow\uparrow} + I^{\downarrow\downarrow} - I^{\uparrow\downarrow} - I^{\downarrow\uparrow})}{(I^{\uparrow\uparrow} + I^{\downarrow\downarrow} + I^{\uparrow\downarrow} + I^{\downarrow\uparrow})} \qquad (1)$$

and plot it for both CAR and ECT as a function of $\theta$ (Fig. 4e). $I^{ij}$ corresponds to the average correlated current $\langle I_{corr} \rangle$ associated with each spin configuration, where $i,j \in \{\uparrow, \downarrow\}$. $C = \pm 1$ when there is a perfect correlation or anti-correlation between the spins of electrons entering the QDs. In contrast, $C = 0$ when the probabilities of equal-spin and opposite-spin transport become equal. When $\mathbf{B} \| \mathbf{B}_{SO}$ we obtain a value of $C = -0.96$ for CAR, demonstrating a nearly perfect singlet pairing between the QDs. Similarly, for ECT $C = +0.93$ is obtained. When $\mathbf{B} \perp \mathbf{B}_{SO}$, $C$ reaches close to 0 for both CAR and ECT, stressing that the triplet component can be tuned to be of similar magnitude to the conventional singlet pairing.

In conclusion, we have used quantum dot-superconductor hybrids to demonstrate highly efficient Cooper pair splitting in a two-dimensional semiconductor platform. Using spin-polarized quantum dots, we performed spin-selective measurements of ECT and CAR and showed that the strong spin-orbit interaction in ternary 2DEGs results in comparable strengths of singlet and triplet correlations between the quantum dots. Finally, through magnetic field rotations, we showed that it is possible to obtain equal amplitudes of ECT and CAR, establishing 2DEGs as an ideal platform to study Majorana bound states in artificial Kitaev chains.

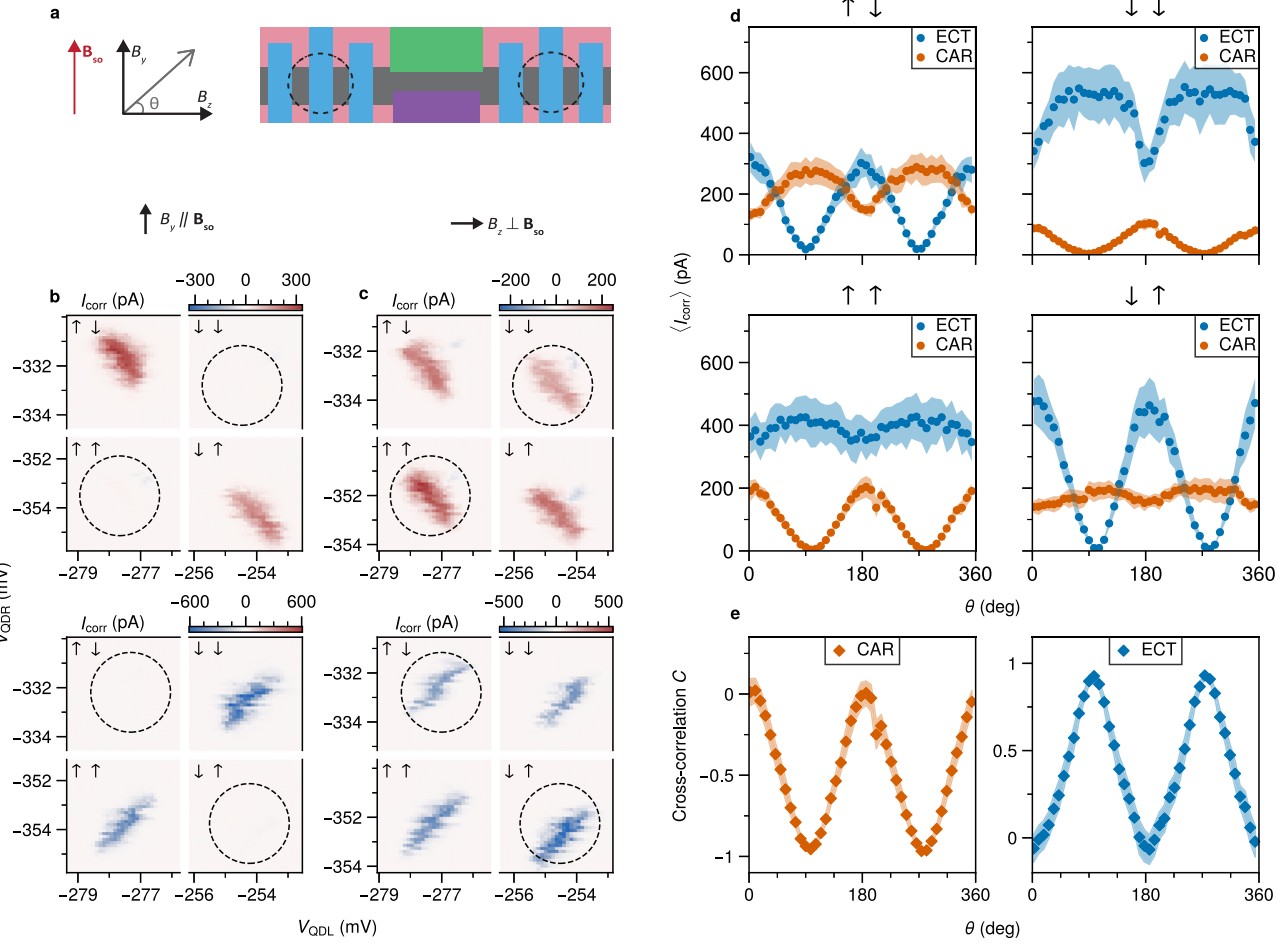

**Fig. 4 | CAR and ECT at finite magnetic field. a** A schematic showing the coordinate system of the applied magnetic field with respect to the device. **b** Measurement of $I_{corr}$ for CAR (top) and ECT (bottom) with $\mathbf{B}\|B_y = 150$ mT and $V_{ABS} = -220$ mV. Lower biases ($|V_L|$, $|V_R| = 70\,\mu V$) are applied to keep the bias window below any sub-gap states, whose energies are pulled down by the finite magnetic field (Fig. S1). Equal-spin CAR and opposite-spin ECT are fully suppressed (circled). **c** Measurement of $I_{corr}$ with $\mathbf{B}\|B_z = 150$ mT. The blockades in (**b**) have been clearly

lifted. **d** Angle-dependence of $\langle I_{corr}\rangle$ for the different spin channels for a full rotation of the magnetic field in the y-z plane. The (−, +) and (+, +) bias configurations are used for ECT and CAR respectively. Each data point represents a single charge stability diagram for a specific spin channel. The data extraction procedure is described in Fig. S7. **e** The calculated spin cross-correlation (as defined in the text) of CAR and ECT, derived from (**d**).

## Discussion

The demonstration of singlet and triplet correlations with Cooper pair splitters in 2DEGs paves the way for more advanced experiments to study entanglement and topological superconductivity. An interesting open question relates to the underlying mechanism that allows for strong triplet CAR in these devices. One possibility is for two equal-spin electrons to form a normal s-wave Cooper pair, due to spin pre-cession in the tunnel barriers. Another path is that an induced p-wave superconducting pairing arises in the hybrid section, such that two equal-spin electrons form a Cooper pair. In order to distinguish these possibilities, we propose to create quasi-1D channels that are bent (rather than straight), resulting in different spin-orbit directions in each arm of the Cooper pair splitter[3,33]. Such devices are easily implemented in 2DEGs where any arbitrary shape of the channel can be realized simply by altering the design of the depletion gates. Given the high fidelity spin correlation we have demonstrated here, such devices could also be used to detect entanglement by performing a Bell test with electrons from a Cooper pair[3].

Finally, the recent realization of a minimal Kitaev chain[18] opens up several possibilities to systematically study Majorana bound states (MBSs). In this regard the 2DEG platform is again particularly suitable. It readily allows for extending these measurements to multi-site QD chains, whereby the flexibility of the 2DEG would allow for the

simultaneous measurement of density of states at the edges and in the bulk. Furthermore, one could use these chains to perform tests of non-Abelian exchange statistics via braiding experiments[34,35], which necessarily require a 2D platform.

## Methods

### Fabrication

Device 1 (main text) and Device 2 (supplementary) were fabricated using techniques described in detail in ref. 36. A narrow aluminum strip is defined in an InSbAs-Al chip by wet etching, followed by the deposition of two normal Ti/Pd contacts. After deposition of 20 nm AlOx via atomic layer deposition (ALD), the two depletion gates are evaporated. Following a second ALD (20 nm AlOx) Ti/Au gates are evaporated in order to define the QDs and tune the ABS energy.

### Measurements

All measurements are performed in a dilution refrigerator with a base temperature of 20 mK. Magnetic fields are applied using a 3D vector magnet. The alignment of the magnetic field with respect to the device is expected to be accurate within ± 5°. Transport measurements are performed in DC using a three-terminal set-up, where the aluminum is electrically grounded (Fig. 1b). Current amplifier offsets are

determined by the average measured current when both dots are in Coulomb blockade. CAR and ECT processes can be observed over a wide range of $V_{ABS}$ voltages. Once a $V_{ABS}$ setting was found with both strong CAR and ECT currents, it was kept at a constant value throughout the rest of the measurements. Further care was taken to implement the same orbitals in both QDs for all presented measurements in the main text. The mismatch between exact $V_{QDR}$ and $V_{QDL}$ values at which ECT and CAR are observed is due to gate instabilities, causing a drift of charge degeneracy points over a period of time. Therefore, the field rotation measurement in Fig. 4e was performed multiple times. No quantitative difference was observed between measurements. Presented data was selected due to high stability of the QDs over the course of the measurements.

Overall, we have measured four fully functional devices at the time of writing this manuscript, all of which have produced highly efficient CAR and ECT mediated by extended ABSs. For three of these devices we have performed magnetic field rotations and observed angle-dependent oscillations of ECT and CAR currents.

## Data availability
Raw data and analysis scripts for all presented figures are available at https://doi.org/10.5281/zenodo.7311374.

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

## Acknowledgements

We thank T. Dvir, G. Wang, C. -X. Liu, M. Wimmer, C. Prosko, P. Makk, R. Aguado, D. Xu, and L. P. Kouwenhoven for valuable discussions and for providing comments on the manuscript. The research at Delft was supported by the Dutch National Science Foundation (NWO) and a TKI grant of the Dutch Topsectoren Program. The work at Purdue was funded by Microsoft Quantum.

## Author contributions

Q.W. and S.L.D.t.H. fabricated and measured the devices. I.K. contributed to the device design and optimization of fabrication flow. MBE growth of the semiconductor heterostructures and the characterization of the materials was performed by D.X., and C.T. under the supervision of M.J.M. The manuscript was written by Q.W., S.L.D.t.h., and S.G., with inputs from all coauthors. S.G. supervised the experimental work in Delft.

## Competing interests

The authors declare no competing interests.
