## [Peer Review File · Nature Communications]

REVIEWER COMMENTS

Reviewer #1 (Remarks to the Author):

The paper by Wang, ten Haaf, Goswami and coworkers reports on Cooper pair splitting in a semiconductor two-dimensional electron gas (2DEG) by coupling two quantum dots to a superconductor-semiconductor hybrid region. Using strong Coulomb interaction in quantum dots the authors have suppressed the local Andreev reflection and reported systematic studies of the elastic cotunneling and the nonlocal crossed Andreev reflection. In the presence of a magnetic field using the spin blockade effect the authors have been able to obtain information about the spin of the electrons forming the Cooper pair. By means of the spin-orbit interaction they have transform the standard spin singlet pairing into equal spin pairing between electrons in two quantum dots.

Despite the fact that the work and the results are quite similar to those from Ref. [16], the results presented in this manuscript have been obtained for the 2DEG system, that can provide some advantages compare to the system of hybrid InSb nanowires studied in Ref. [16]. The results presented in Fig.4(g) are especially important because they demonstrate a high fidelity spin correlation that varies between 0% and -96% while in Fig.ED8 in Ref. [16] form -47% to -86%. While the manuscript doesn't report on entanglement, it achieves an important milestone by demonstrating this high fidelity spin correlation measurement.

The data is clear and the analysis is performed in a careful manner. The quality and the significance of the obtained results are high enough therefore I believe that the work is suitable for Nature Communication.

Before I would like to ask the authors to comment on the following points:

1. It is not clear how $\langle I_{\text{corr}} \rangle$ is obtained. How is the average calculated?
2. Above the line 268 in the definition of C. There is a lack of definition of $I^{\uparrow\uparrow}$ etc. Does it mean $\langle I_{\text{corr}}^{\uparrow\uparrow} \rangle$?
3. Line 227. I am afraid that the magnetic field of 35 mT is too weak to polarize the nuclear spins at $T = 20$ mK.
4. There is a question about tunability of B_{SO} in the experiment. Is it possible in the current experiment? It would be interesting to check how it would affect the effect under study.
5. The authors claim "the exceptionally strong spin-orbit interaction in our 2DEGs". It could be useful to estimate the B_{SO} amplitude and compare it with that from Ref.[16].

6. In Fig.3(a), the two black down arrows depicted two electrons with down spin should perhaps be placed in the dots not in the leads. Same in (b) and (e).

7. It could be useful to explain in the paper the mechanism - how “the large spin-orbit coupling in our 2DEGs leads to strong equal-spin CAR currents that are of similar amplitude to the conventional opposite-spin processes.”

8. The scheme from Fig.4(d) could be misleading - the effect of spin-orbit interaction on the split Cooper pair state (two black slanted opposite arrows). Due to spin orbit interaction both spins precess in a different way leading to state like $|\uparrow\rightarrow\rangle$. Formally, for the singlet state $|S\rangle = |\uparrow\downarrow\rangle - |\downarrow\uparrow\rangle$ we can rotate the left spin into $|\uparrow\rangle \rightarrow a|\uparrow\rangle + b|\downarrow\rangle$ and $|\downarrow\rangle \rightarrow b|\uparrow\rangle + a|\downarrow\rangle$. Then $|S\rangle \rightarrow a|\uparrow\downarrow\rangle + b|\downarrow\downarrow\rangle - a|\downarrow\uparrow\rangle - b|\uparrow\uparrow\rangle = a(|\uparrow\downarrow\rangle - |\downarrow\uparrow\rangle) + b(|\downarrow\downarrow\rangle + |\uparrow\uparrow\rangle)$. Therefor depending on parameters a, b we can rotate the single state into the equal spin state. See e.g. the discussion and Eq.(25) from Tomaszewski et al. Phys. Rev. B 98, 174504 (2018).

9. Since the paper is similar to Ref. [16] it may be useful to briefly discuss the differences and advances of the new experiment.

Reviewer #2 (Remarks to the Author):

The manuscript reports excellent data from a Cooper pair splitter device based on a 2D semiconductor. The latter is in my opinion the most significant achievement of this work and will allow to go to more complex geometries and tunable coupling in future experiments. In spite of several fundamental and smaller issues I list below, I expect that this work will be central to a new family of CPS and other experiments that would be much more problematic in nanowire based structures. I therefore support the publication of this manuscript in Nature Communications, provided that all points below can be addressed satisfactorily.

Fundamental issues:

- The manuscript is almost identical in structure and analysis to Ref. [16]. This is a bit unfortunate, but I believe that the authors could make a point (and a big difference) by providing a thorough, systematic and critical discussion of each element of the experiment, especially about the triplet superconductivity part and the eigenstates of the detector QDs (see more on this below).

- CPS has been shown in 2d systems (graphene, Ref. [15]) and in semiconducting systems (many references). The authors bring forward the argument of scaling up and geometrical constraints as a motivation in the introduction, but I feel the paper would benefit from more specific ideas, like designed interactions in an array of gate defined QDs (there are probably other even more interesting ideas

around). Or put in another way: why are Kitaev chains interesting if there is already p-wave pairing? Since this is the main point that makes a difference to Ref. [16], it would make sense to polish this point a bit more and contrast this system to nanowire based ones to warrant the publication in Nature Comm.

- The authors implicitly assume a strong tunnel coupling between the superconductor and the semiconductor below it (proximity effect), and simultaneously a strong spin-orbit interaction (SOI) (thus resulting in a triplet pairing part). This suggests that SOI is very strong in this material, as has been demonstrated before. At the same time, it is assumed that SOI is negligible in the QDs made of the same host material, because the authors only consider pure spin eigenstates in their explanations. I would expect quite the opposite (reduced SOI below the superconductor due to strong electron exchange with the aluminum), or at the very least similar magnitudes. In this case the QD eigenstates are spin-orbit eigenstates and not pure spin eigenstates, so that the measurement corresponds to a projection onto these eigenstates with a “rotated” spin part. That this is the case is evident from the curvature in the B-field dependence of the resonances and from the non-paired spin filling in S6.

While the authors suggest an effect of SOI on the Cooper pairing (-> triplet part), and tentatively discuss a spin rotation in the tunnel barriers or the interface to the superconductor (too short? SOI is strong...), they (and Ref. [16]) should discuss the effects of a SOI-based spin rotation in the detector QDs – an attempt was made in Ref. [30], though in the present case SOI seems even larger and the argument might not hold here, which naturally would result in current components that erroneously would be attributed to parallel spin currents.

I would like to stress that this issue does not belittle the beauty of the experiment, nor the CPS part, but questions the interpretation of a triplet proximity region.

- To conclude the above argument, this spin rotation due to SOI on the QDs might be strongly anisotropic (crystal structure, g-factor anisotropy, device geometry, others?) and depend on the orientation of the magnetic field, which would be an alternative explanation of the dependence on the external field direction, and naturally would account for the lifting of the Pauli blockade observed in the first part of the paper. This would make the interpretation as triplet pairing implausible.

- If the authors then come to the conclusion and carefully argue that there really is a triplet part in the pairing function, they should then explain in addition why there seems to be no Majorana bound states one might expect for topologically non-trivial proximity regions (zero-bias peak in conductance, others?). And if there were such states, what would be the signature in the CPS experiment? I believe there should be theoretical works on this question published already.

- The authors should show experiments in which they used different resonances, for example the next neighboring ones (not just two selected pairs), if they have such data available. This would increase the credibility of the correct spin detection considerably, including the spin detection by the QDs.

- Other papers report different CPS signatures due to the charge dynamics between the QDs. Since the linewidths here are very narrow (allowing to obtain the near ideal CPS efficiencies), this should be addressed (though I believe that this is not a problem here, since all signals seem to add up perfectly well.) If correct, this suggests that there is no single particle transport in the proximity region and that the complete semiconductor region below the superconductor is proximitized.

But then the argumentation of the double QD spin states is not obvious to me:

First, the argument that the independent transitions $1 \leftrightarrow 2$ of each QD is only allowed for «opposite spin» (opposite to the one already on the QD) is only applicable with an exchange splitting on each individual QD, and of the same sign on each QD. Otherwise, these states are degenerate and no spin filtering / spin blockade occurs, or might act as opposite spin filters (\rightarrow opposite to each other, which is what is required here). Second, the double QD Pauli blockade without a superconductor requires a direct tunnel coupling between the QDs, which is not obvious here due to the proximity region. How the superconductor renormalizes the double QD states seems not exactly trivial, but one might expect only weak effects due to the strong barriers.

So if we take Fig. 3 with $B=0$ as a demonstration of a Pauli blockade between the QDs, with the still visible transitions for the “blocked” states as the leakage currents, or the error in the spin state (S/T) measurement ($\sim 50\%$) – with purely s-wave Cooper pairs as a “reference”, the leakage might simply be due to SOI eigenstates.

Small points:

- Abstract, logic in sentence: Cooper pairs form not only in s-wave, but probably in all superconductors (but possibly with different wavefunction symmetries)
- The manuscript does not actually demonstrate new physics due to the 2D nature of the host material (opposed to the graphene reference). If possible, the authors could speculate whether there is 2D physics to be explored here as well.
- The term “virtual state” is rather unfortunate (often used if two subsystems are coupled and the eigenstates attributed (wrongly) solely to one). Here I believe one should speak of higher order processes (in the tunneling hamiltonian).
- Bell test is discussed in Ref. [29] as well and put in a broader context.
- Schematic: symbol seems to be an ac source (content clear, though)
- «Irregular Coulomb diamonds ... indicates well-defined orbital levels» makes little sense.
- Weak coupling to proximity region, not to the superconductor
- Line 120: wrong reference? E.g. Gramich et al., PRL 115, 216801 (2015) also discuss the shift in gate voltage observed here.

- QDR shows «Coulomb shards» suggesting 2 QDs in series to the right, and potentially a smaller voltage applied to SC.
- Line 168: wrong reference, should be [14]
- Line 177: cite Ref [16]
- Schematic arrows in reservoirs might be mistaken as spin polarized reservoirs
- Line 267: this equation is from Ref. [30], not [16], where it is only applied
- Is 150mT out of plane no problem for Al, even if thin? -> quasi-particle poisoning might become relevant.
- Are the resonances in S6 identified correctly? Gate voltage for left QD seems to be off by half a period. Which ones were used in the main text?
- Line 294: cite Ref. [3] here and corresponding experiments by Hels et al. Phys. Rev. Lett. 117, 276802 (2016)

Reviewer #3 (Remarks to the Author):

In the manuscript Wang et al. demonstrate Cooper pair splitting in a 2DEG for the first time. Beside high splitting efficiency they demonstrate a large tunability of spins of the split electron pair by changing the direction of the external magnetic field. The presented results demonstrate well the suitability of InAsSb 2DEGs for quantum electronics applications. I find their work interesting and timely. The manuscript is well written and suitable for publishing in Nature Communications. However, I suggest the authors to consider addressing the following points in the published version of the manuscript.

Line 226: I believe that 35 mT does not polarize the nuclear spins, but it produces a large enough Zeeman splitting (for the electrons) that the hyperfine field cannot mix the two spin species anymore.

i) In the captions of Figs. 2,3&4 the authors indicate that voltage on the central 'ABS gate' was -245 and -220 mV. However on Fig. 1 they only present the tunnel spectrum of the ABS states above -150 mV. I miss further details on ABS at the gate voltages used for the splitting measurements (e.g. What is its energy? How much is it electron/hole-like?) Did the authors see the tunability of the CAR/ECT ratio with V_{ABS} (similar to Bordin et al. arxiv.org/abs/2212.02274)?

ii) The large difference of the g-factors of the dots indicates that the details of the confinement have a large impact on the g-tensors. How anisotropic are the g-tensors of the two dots? Knowing the anisotropy, how large is the angle enclosed by the spin quantization axes?

iii) For many years the demonstration of high splitting efficiency was lacking due to the large coupling induced level broadening of the dots. I think it would be beneficial if the authors could comment on the tunnel couplings they achieved in the presented work.

The reviews comments are in blue, our response in black and changes to the manuscripts are in red

We want to thank all three referees for their constructive comments. Please find below a point by point response. Please note that all references cited in this response correspond to the resubmitted version of the manuscript.

Reviewer #1 (Remarks to the Author):

The paper by Wang, ten Haaf, Goswami and coworkers reports on Cooper pair splitting in a semiconductor two-dimensional electron gas (2DEG) by coupling two quantum dots to a superconductor-semiconductor hybrid region. Using strong Coulomb interaction in quantum dots the authors have suppressed the local Andreev reflection and reported systematic studies of the elastic co-tunneling and the nonlocal crossed Andreev reflection. In the presence of a magnetic field using the spin blockade effect the authors have been able to obtain information about the spin of the electrons forming the Cooper pair. By means of the spin-orbit interaction they have transform the standard spin singlet pairing into equal spin pairing between electrons in two quantum dots.

Despite the fact that the work and the results are quite similar to those from Ref. [16], the results presented in this manuscript have been obtained for the 2DEG system, that can provide some advantages compare to the system of hybrid InSb nanowires studied in Ref. [16]. The results presented in Fig.4(g) are especially important because they demonstrate a high fidelity spin correlation that varies between 0% and -96% while in Fig.ED8 in Ref. [16] form -47% to -86%. While the manuscript doesn't report on entanglement, it achieves an important milestone by demonstrating this high fidelity spin correlation measurement.

The data is clear and the analysis is performed in a careful manner. The quality and the significance of the obtained results are high enough therefore I believe that the work is suitable for Nature Communication.

We thank the referee for the positive assessment of our experimental results and for highlighting the versatility of a semiconductor 2DEGs platform in comparison to hybrid nanowires.

Before I would like to ask the authors to comment on the following points:

1. It is not clear how is obtained. How is the average calculated?

The $\langle I_{\text{corr}} \rangle$ values in Fig.4(f) are extracted from the measured charge stability diagrams at each specified angle. The raw data from these measurements along with the data processing methodology is detailed in supplementary Fig. S7.

2. Above the line 268 in the definition of C. There is a lack of definition of $I^{\uparrow\uparrow}$ etc. Does it mean ?

It stands for $\langle I_{\text{corr}} \rangle$ extracted for the $\uparrow\uparrow$ -spin configuration of the quantum dots. We added another sentence to clarify this:

I^{ij} corresponds to the average correlated current $\langle I_{\text{corr}} \rangle$ associated with the spin configuration, in which $i, j \in \{\uparrow, \downarrow\}$.

3. Line 227. I am afraid that the magnetic field of 35 mT is too weak to polarize the nuclear spins at T = 20 mK.

Indeed, this is correct. What we intended to convey is that a field of 35 mT results in a Zeeman splitting that is larger than effective broadening due to the hyperfine interaction. Therefore, the spins in the QDs are well defined. We replace the sentence with:

We confirm this by applying a magnetic field to suppress spin-mixing due to the hyperfine interaction and find that 35mT is sufficient to fully suppress the remaining current (Fig.S5).

4. There is a question about tunability of B_SO in the experiment. Is it possible in the current experiment? It would be interesting to check how it would affect the effect under study.

The amplitude of B_SO in the hybrid section could in principle be tuned by electric field. However, as described in our response to Comment 5 (below), quantitatively extracting B_SO requires several assumptions that may not be valid in our system.

5. The authors claim “the exceptionally strong spin-orbit interaction in our 2DEGs”. It could be useful to estimate the B_SO amplitude and compare it with that from Ref.[16].

From studying weak anti-localization (WAL) in these 2DEGs, we estimated a Rashba parameter α of about $0.2\sim 0.3 \text{ meV}\text{\AA}$ (ref. [22]). Proximity to the superconductor can lead to renormalization of α . In ref. [17] (used to be ref[16]), SOC strength in the hybrid segment is extracted using a model that is only valid in the weak spin-orbit coupling limit. In particular in this model it is expected that the spin-filtered current should vary sinusoidally with field angle. Unlike ref.[17], our data (see Fig. 4f) shows clear deviations from this, indicating that the model may not be applicable. We therefore decided not to include this analysis in our manuscript.

Nevertheless, if we use the same procedure as described in ref. [17], we get a Rashba parameter α in between $0.09 \text{ eV}\text{\AA}$ and $0.12 \text{ eV}\text{\AA}$ (see figure below)

6. In Fig.3(a), the two black down arrows depicted two electrons with down spin should perhaps be placed in the dots not in the leads. Same in (b) and (e).

We thank the referee for the suggestions and modified the schematics as advised.

Fig. 3abe: moved the black arrows from the leads to the dots

7. It could be useful to explain in the paper the mechanism - how “the large spin-orbit coupling in our 2DEGs leads to strong equal-spin CAR currents that are of similar amplitude to the conventional opposite-spin processes.”

We further clarify our statements by replacing the above-mentioned sentence with:

The large spin-orbit coupling in our 2DEGs, in combination with the device length, results in significant spin precession of the electrons.

Importantly, we show that this leads to strong equal-spin CAR currents that are of similar amplitude to the conventional opposite-spin processes.

8. The scheme from Fig.4(d) could be misleading - the effect of spin-orbit interaction on the split Cooper pair state (two black slanted opposite arrows). Due to spin orbit interaction both spins precess in a different way leading to state like $|\uparrow \rightarrow \rangle$. Formally, for the singlet state $|S\rangle = \frac{1}{\sqrt{2}}(|\uparrow \downarrow\rangle - |\downarrow \uparrow\rangle)$ we can rotate the left spin into $|\uparrow\rangle \rightarrow a|\uparrow\rangle + b|\downarrow\rangle$ and $|\downarrow\rangle \rightarrow b|\uparrow\rangle + a|\downarrow\rangle$. Then $|S\rangle \rightarrow a|\uparrow \downarrow\rangle + b|\downarrow \downarrow\rangle - a|\downarrow \uparrow\rangle - b|\uparrow \uparrow\rangle = a(|\uparrow \downarrow\rangle - |\downarrow \uparrow\rangle) + b(|\downarrow \downarrow\rangle - |\uparrow \uparrow\rangle)$

$|\downarrow\rangle - |\downarrow \uparrow\rangle + b(|\downarrow \downarrow\rangle + |\uparrow \uparrow\rangle)$. Therefore depending on parameters a, b we can rotate the single state into the equal spin state. See e.g. the discussion and Eq.(25) from Tomaszewski et al. *Phys. Rev. B* 98, 174504 (2018).

We thank the referee for pointing out this reference and the relevant discussions. We feel that the explanation of the data is actually complete even without these schematics.

We have therefore removed these schematics, and appropriately modified the figure labels in the text.

9. Since the paper is similar to Ref. [16] it may be useful to briefly discuss the differences and advances of the new experiment.

The realization of similar physics (high efficient CPS, triplet components) on our platform is, to our opinion, laying down the foundation of many potentially interesting experiments. We already proposed a few of these in the previous “Discussion” section.

We now extend the discussion section by providing more concrete examples.

Reviewer #2 (Remarks to the Author):

The manuscript reports excellent data from a Cooper pair splitter device based on a 2D semiconductor. The latter is in my opinion the most significant achievement of this work and will allow to go to more complex geometries and tunable coupling in future experiments. In spite of several fundamental and smaller issues I list below, I expect that this work will be central to a new family of CPS and other experiments that would be much more problematic in nanowire based structures. I therefore support the publication of this manuscript in Nature Communications, provided that all points below can be addressed satisfactorily.

We thank the referee for appreciating the significance of this work in the CPS community and the potential of the our hybrid 2D semiconductor platform.

Fundamental issues:

- The manuscript is almost identical in structure and analysis to Ref. [16]. This is a bit unfortunate, but I believe that the authors could make a point (and a big difference) by providing a thorough, systematic and critical discussion of each element of the experiment, especially about the triplet superconductivity part and the eigenstates of the detector QDs (see more on this below).

- CPS has been shown in 2d systems (graphene, Ref. [15]) and in semiconducting systems (many references). The authors bring forward the argument of scaling up and geometrical constraints as a motivation in the introduction, but I feel the paper would benefit from more specific ideas, like designed interactions in an array of gate defined QDs (there are probably other even more interesting ideas around).

We agree with the importance of explicitly outlining the ideas ready to be implemented in our platform.

In order to make this clearer, we have now extended the Discussion section with some concrete examples.

Or put in another way: why are Kitaev chains interesting if there is already p-wave pairing? Since this is the main point that makes a difference to Ref. [16], it would make sense to polish this point a bit more and contrast this system to nanowire based ones to warrant the publication in Nature Comm.

We want to clarify that we have been careful to not claim p-wave pairing in the hybrid region. Perhaps the source of confusion arises from our use of the term “p-wave superconducting pairing” in the Discussion section. Our observations demonstrate a pairing between two equal-spin states (i.e., triplet pairing) on the two quantum dots. As we have mentioned in the Discussion section, the origin of this pairing is difficult to ascertain from the current experiments (and also from those in nanowires).

With regards to this specific paper, the main contrast with nanowires is with respect to the platform itself. We believe that the 2D platform allows for more advanced experiments (as further detailed in the Discussion section).

- The authors implicitly assume a strong tunnel coupling between the superconductor and the semiconductor below it (proximity effect), and simultaneously a strong spin-orbit interaction (SOI) (thus resulting in a triplet pairing part). This suggests that SOI is very strong in this material, as has been demonstrated before.

At the same time, it is assumed that SOI is negligible in the QDs made of the same host material, because the authors only consider pure spin eigenstates in their explanations.

I would expect quite the opposite (reduced SOI below the superconductor due to strong electron exchange with the aluminum), or at the very least similar magnitudes. In this case the QD eigenstates are spin-orbit eigenstates and not pure spin eigenstates, so that the measurement corresponds to a projection onto these eigenstates with a “rotated” spin part. That this is the case is evident from the curvature in the B-field dependence of the resonances and from the non-paired spin filling in S6.

In order to operate these QDs as charge and spin filters, we specifically select a pair of resonances that corresponds to the filling of a single orbital—where they are separated by only charging energy at zero field and show linear Zeeman splitting with small magnetic field. At higher fields the Zeeman energy becomes larger than the orbital level spacing, resulting in the apparent curvature (for example at around $B = 0.55\text{T}$ and $V_{QDL} \approx -235\text{ mV}$ in Fig.S6(a)). We make sure to operate at fields well below this field.

While the authors suggest an effect of SOI on the Cooper pairing (\rightarrow triplet part), and tentatively discuss a spin rotation in the tunnel barriers or the interface to the superconductor (too short? SOI is strong...), they (and Ref. [16]) should discuss the effects of a SOI-based spin rotation in the detector QDs – an attempt was made in Ref. [30], though in the present case SOI seems even larger and the argument might not hold here, which naturally would result in current components that erroneously would be attributed to parallel spin currents.

I would like to stress that this issue does not belittle the beauty of the experiment, nor the CPS part, but questions the interpretation of a triplet proximity region.

The SOC eigenstates within the QD could indeed produce similar experimental observations as shown here. In ref. [17] they performed dot spectroscopy measurements (Fig. ED3) to qualitatively estimate the contribution of these admixtures to the equal-spin CAR current. While such measurements were not performed on the specific devices in the manuscript, following the referee’s suggestions we have now done these experiments on a similar device with the same material and a similar QD design. We have measured the following Coulomb diamonds and excited state spectra:

where B_z is the field direction along the channel. The anti-crossing is due to SOC and gives rise to the spin-orbital repulsion gap $2\langle H_{SO} \rangle$. The amount of the spin-admixture is $\langle H_{SO} \rangle / (\delta - E_z)$ according to first-order perturbation theory (Hanson et al., *RMP*, 2007) where δ is the level spacing and E_z the Zeeman splitting—both these quantities can be read out directly from the measurement. Thus, the opposite-admixture within one QD is less than $0.28/1.68 = 16.6\%$ at $B_z = 150\text{mT}$. If one follows the estimation procedure as detailed in ref.[17], the expected triplet component at $B \parallel B_z$ should be no more than 10.8% assuming no SOC in the superconducting hybrid. In our experiments the observed triplet-singlet ratio at these angles is in between 62% to 143%, thus the majority of the contribution to the triplet component should originate from the SOC in the hybrid. We note that these energy scales (and thus the estimate above) can depend on the exact details of the QD and thus the results presented here are intended to be used solely as a broad approximation.

From a different perspective, we have observed that for two different combinations of QD orbitals (one in Fig.4 and another one in Fig.S6), the spin-filtered CAR and ECT currents show similar oscillation behaviors with the same suppression angles. As the spin-orbit direction and amplitude can vary from one orbital to another, it is very unlikely that the observation can be explained only by the SOC eigenstates in QDs.

- To conclude the above argument, this spin rotation due to SOI on the QDs might be strongly anisotropic (crystal structure, g-factor anisotropy, device geometry, others?) and depend on the orientation of the magnetic field, which would be an alternative explanation of the dependence on the external field direction, and naturally would account for the lifting of the Pauli blockade observed in the first part of the paper. This would make the interpretation as triplet pairing implausible.

We agree that the SOI in the QDs could in principle give rise to the lifting of the blockade at higher fields, as discussed above. We will comment on the suggested explanations for the dependence on the external field direction:

g-factor anisotropy:

We indeed have a slightly anisotropic g -factor in QDs. The magnetic field dependence of the left quantum dot for device 2 gives rise to g -factors of about -18 for B_z (in-plane, parallel to the channel) and -23 for B_y (in-plane, perpendicular to channel, see below).

However, as argued in [Nadj-Perge et al., *PRL*, 2012], the g -factor anisotropies most likely do not give rise to the observed current anisotropy. In a nutshell, when the spin-quantization axis in the two dots are not parallel, there will two in-plane directions where spin-flipping processes are fully blocked. Therefore, the measured current would encounter four peaks dips in a full-angle 360° rotation, inconsistent with our observed periodicity.

Crystalline anisotropy and device geometry:

We have obtained similar results on different devices and even on different chips, where every device are slightly different from the others (Al etch, width of depletion gates etc.). Thus, we don't think device geometry is the main cause of the such an observation.

- If the authors then come to the conclusion and carefully argue that there really is a triplet part in the pairing function, they should then explain in addition why there seems to be no Majorana bound states one might expect for topologically non-trivial proximity regions (zero-bias peak in conductance, others?). And if there were such states, what would be the signature in the CPS experiment? I believe there should be theoretical works on this question published already.

We would like to stress again that we cannot draw conclusion about a triplet part in the pairing function within the hybrid section, but only between equal-spin states on the QDs. It should furthermore be noted that the hybrid section in these devices are very short (~ 200 nm) such that even if a topological

phase transition were to occur, the wavefunctions of two MZMs would strongly overlap and therefore one would not expect to see persisting zero-bias peaks.

- The authors should show experiments in which they used different resonances, for example the next neighboring ones (not just two selected pairs), if they have such data available. This would increase the credibility of the correct spin detection considerably, including the spin detection by the QDs.

The measurement presented in Fig. S6 are done with different pairs of resonances in comparison to the main text (labelled with dashed line). **We add another legend on top of Fig. S6a to clarify solid and dash lines within S6 (a,b)**

- Other papers report different CPS signatures due to the charge dynamics between the QDs. Since the linewidths here are very narrow (allowing to obtain the near ideal CPS efficiencies), this should be addressed (though I believe that this is not a problem here, since all signals seem to add up perfectly well.)

this suggests that there is no single particle transport in the proximity region and that the complete semiconductor region below the superconductor is proximitized.

The referee is correct about abovementioned points.

But then the argumentation of the double QD spin states is not obvious to me:

First, the argument that the independent transitions $1 \leftrightarrow 2$ of each QD is only allowed for «opposite spin» (opposite to the one already on the QD) is only applicable with an exchange splitting on each individual QD, and of the same sign on each QD. Otherwise, these states are degenerate and no spin filtering / spin blockade occurs, or might act as opposite spin filters (\rightarrow opposite to each other, which is what is required here).

We observe clear even-odd diamonds and linear Zeeman dependence of Coulomb resonances. This indicates that our QDs have large effective level spacing (due to the small effective mass in our case). Therefore, there is no orbital degeneracy and Pauli blockade forces the filling of opposite-spin electron at the $1 \leftrightarrow 2$ transition for each QD.

Second, the double QD Pauli blockade without a superconductor requires a direct tunnel coupling between the QDs, which is not obvious here due to the proximity region. How the superconductor renormalizes the double QD states seems not exactly trivial, but one might expect only weak effects due to the strong barriers.

Indeed due to the narrow QD linewidth, hard superconducting gap and weak coupling between the QDs and hybrid section, only ECT and CAR are responsible for the transport between two QDs.

So if we take Fig. 3 with $B=0$ as a demonstration of a Pauli blockade between the QDs, with the still visible transitions for the “blocked” states as the leakage currents, or the error in the spin state (S/T) measurement ($\sim 50\%$) – with purely s -wave Cooper pairs as a “reference”, the leakage might simply be due to SOI eigenstates.

It is unlikely that SOC lift the Pauli blockade at zero field and this has been discussed in literature, for example ref.[30]. Intuitively speaking, at zero field electrons with any arbitrary spin orientations can tunnel from the normal leads to the QDs with a certain amount of spin precession. Thus there would be

a situation where the combination of spins violate the spin-conservation (equal spin for ECT, opposite spin for CAR). This disrupts the transport cycles and thus results in blocked current.

Besides, the observed zero-field leakage current is suppressed with only a small magnetic field (~35 mT). Thus, it is attributed to the hyperfine interaction.

Small points:

- Abstract, logic in sentence: Cooper pairs form not only in s-wave, but probably in all superconductors (but possibly with different wavefunction symmetries)

Correct. We modify the sentence as:

Cooper pairs occupy the ground state of superconductors and are typically composed of maximally entangled electrons with opposite spin.

- The manuscript does not actually demonstrate new physics due to the 2D nature of the host material (opposed to the graphene reference). If possible, the authors could speculate whether there is 2D physics to be explored here as well.

As we have commented earlier, in this manuscript we aim to demonstrate the similar physics on this new 2D platform, which allows for more advanced experiments (more in Discussion section).

- The term “virtual state” is rather unfortunate (often used if two subsystems are coupled and the eigenstates attributed (wrongly) solely to one). Here I believe one should speak of higher order processes (in the tunneling hamiltonian).

Line 18: we replace the term “virtually state” with “higher order process”

- Bell test is discussed in Ref. [29] as well and put in a broader context.

Cite this reference now also in the Introduction section—the new citation number is [6].

Schematic: symbol seems to be an ac source (content clear, though)

The voltage sources within Fig.1b are replaced with the standard DC source symbols.

- «Irregular Coulomb diamonds ... indicates well-defined orbital levels» makes little sense.

We modify the relevant sentences:

The charge stability diagrams of both QDs show Coulomb diamonds with clear even-odd spacing. The pair of Coulomb peaks show linear splitting as a function of magnetic field, indicative of a spin-degenerate single orbital level (Fig.S1).

- Weak coupling to proximity region, not to the superconductor

True. Line 128 now with “...a weak coupling to the proximity region”

- Line 120: wrong reference? E.g. Gramich et al., PRL 115, 216801 (2015) also discuss the shift in gate voltage observed here.

both [R.S. Deacon et. al, PRB, 2010] and [Gramich et.al, PRL 2015] are cited at line 129.

- QDR shows «Coulomb shards» suggesting 2 QDs in series to the right, and potentially a smaller voltage applied to SC.

The induced gap is of similar size for the left and right dot diamond.

- Line 168: wrong reference, should be [14]

Both [15] (Schindele et.al, PRL, 2012) and [17] (Wang et. al., Nature, 2022) are now cited, because we are using the adapted definition of CPS efficiency from [17].

- Line 177: cite Ref [16]

Reference added

- Schematic arrows in reservoirs might be mistaken as spin polarized reservoirs

Fig. 3abe: move the black arrows from the leads to the dots

- Line 267: this equation is from Ref. [30], not [16], where it is only applied

Ref. [17] is removed

- Is 150mT out of plane no problem for Al, even if thin? -> quasi-particle poisoning might become relevant.

The 150 mT is only applied in-plane, which is small compared to the in-plane critical field of Al (evident from Fig. S1). No measurements were performed with out of plane fields.

- Are the resonances in S6 identified correctly? Gate voltage for left QD seems to be off by half a period. Which ones were used in the main text?

Yes they are labelled correctly. However, the quantum dot levels drift slowly over time and thus a perfect match between the S6 and other measurements shown in the main test is hard. While we are doing the measurement we track the Coulomb resonances over time.

- Line 294: cite Ref. [3] here and corresponding experiments by Hels et al. Phys. Rev. Lett. 117, 276802 (2016)

These two references are now cited.

Reviewer #3 (Remarks to the Author):

In the manuscript Wang et al. demonstrate Cooper pair splitting in a 2DEG for the first time. Beside high splitting efficiency they demonstrate a large tunability of spins of the split electron pair by changing the direction of the external magnetic field. The presented results demonstrate well the suitability of InAsSb 2DEGs for quantum electronics applications. I find their work interesting and timely. The manuscript is well written and suitable for publishing in Nature Communications. However, I suggest the authors to consider addressing the following points in the published version of the manuscript.

We thank the referee for the positive assessment.

Line 226: I believe that 35 mT does not polarize the nuclear spins, but it produces a large enough Zeeman splitting (for the electrons) that the hyperfine field cannot mix the two spin species anymore.

Indeed, we modified relevant texts and thank the referee for the suggestion.

i) In the captions of Figs. 2,3&4 the authors indicate that voltage on the central 'ABS gate' was -245 and -220 mV. However on Fig. 1 they only present the tunnel spectrum of the ABS states above -150 mV. I miss further details on ABS at the gate voltages used for the splitting measurements (e.g. What is its energy? How much is it electron/hole-like?)

Measurements with dots require applying voltages to four additional finger gates, in comparison to tunneling spectroscopy. Due to the capacitance cross-talk, we cannot exactly identify the ABS states in spectroscopy which are mediating the transport in Figs 2,3&4.

What we can infer is that the relevant ABS states have an energy larger than $120 \mu\text{V}$ at zero field (applied bias voltages) due to the absence of local signals (example seen Fig.S3). We cannot however resolve the charge characters of the ABS states from our results.

Did the authors see the tunability of the CAR/ECT ratio with V_{ABS} (similar to Bordin et al. arxiv.org/abs/2212.02274)?

We have observed similar tunability of amplitudes of CAR and ECT as a function of V_{ABS} on device 2, data shown below (top panel tunneling spectroscopy, bottom panel extracted CAR/ECT current).

One can associate these measurements with the prediction in theory (ref. [18]) and the experimental results (Bordin et al). Some features, like the weak side-peaks of ECT signal around the gate voltage with minimal ABS energy, don't match expectation from the simple theory model with a single ABS. However, since we did not perform similar measurements on device 1 and this is not our main focus of this manuscript, we did not include this in our paper.

ii) The large difference of the g -factors of the dots indicates that the details of the confinement have a large impact on the g -tensors. How anisotropic are the g -tensors of the two dots? Knowing the anisotropy, how large is the angle enclosed by the spin quantization axes?

We have anisotropic g -factor (see response for referee 2 with subtitle “ **g -factor anisotropy:**”).

However, we have not performed the angle-rotation measurements to extract the g -tensor and thus are not able to further investigate the spin quantization axes based on g -factor anisotropy.

iii) For many years the demonstration of high splitting efficiency was lacking due to the large coupling induced level broadening of the dots. I think it would be beneficial if the authors could comment on the tunnel couplings they achieved in the presented work.

We attribute the high CPS efficiency obtained here to the narrow bandwidth of the QDs and the hard induced superconducting gap. To extract the dot-lead coupling and effective coupling we use the theory model in ref. [19].

CAR/ECT current measured in $\epsilon_L - \epsilon_R$ plane has the forms:

$$I_{CAR} = \frac{e}{\hbar} \cdot \frac{\Gamma_{DL}^2}{\Gamma_{DL}^2 + (\epsilon_L + \epsilon_R)^2} \cdot \frac{|\Gamma^{CAR}|^2}{\Gamma_{DL}}$$

$$I_{ECT} = \frac{e}{\hbar} \cdot \frac{\Gamma_{DL}^2}{\Gamma_{DL}^2 + (\epsilon_L - \epsilon_R)^2} \cdot \frac{|\Gamma^{ECT}|^2}{\Gamma_{DL}}$$

with Γ_{DL} being the total QD-lead coupling strength (i.e. sum of left QD-left N and right QD-right N), Γ^{CAR} and Γ^{ECT} the effective coupling between the two QDs. Plotted in the $\epsilon_L - \epsilon_R$ plane, the current has a Breit-Wigner form with the broadening being Γ_{DL} and reaching the maximum value along $\epsilon_L \mp \epsilon_R$ for CAR and ECT, respectively.

Fitting the measurements shown in Fig.2 (e,f) to the equation, we obtain a total dot-lead coupling of about **30-40 μeV** and effective CAR/ECT coupling about **2-3 μeV** (see below).

(a Savitzky-Golay filter of window length 3 and polynomial order 1 is applied to the measured current before the fitting)

REVIEWERS' COMMENTS

Reviewer #1 (Remarks to the Author):

The resubmitted paper by Wang, ten Haaf, Goswami and coworkers incorporates the comments by the reviewers in a way that makes the text more accessible. I feel that the revised manuscript has further improved. I am in favor of publication. Just one minor point: I would suggest to add a short note to the main manuscript that the information about how the $\langle I_{\text{corr}} \rangle$ values are extracted and the data processing methodology are detailed in supplementary Fig. S7.

Reviewer #2 (Remarks to the Author):

The picture the authors seem to have – which is very plausible to me – is that the superconductor and the hybrid region is a near-perfect source of singlet electron pairs that then get transformed by spin orbit coupling (SOC) into triplets in the non-proximitized semiconductor region, and then detected in the Zeeman polarized QDs. I fully agree with this view, with the small question mark of why the effects of SOC should be stronger in the shorter semiconductor region than in the longer QD regions. But I am ok with this, since it is clearly described how this conclusion was reached. All minor points I made before were addressed, too.

This simple view could very easily be illustrated schematically in Fig. 1, with a singlet CPS pair, a subsequent spin rotation and the spin detection on the QDs. Maybe one can even put bra-kets to label the singlets and triplets at the corresponding positions.

My stumbling block was and still is that the wording in the abstract and in the introduction seems to suggest that there is a triplet pairing (these are the words), which the authors use in the sense of a triplet correlation or as triplet states between the QDs, but not in the sense of a superconducting triplet pairing. Most of my main concerns came from this use of the word “pairing” that in my view should be reserved for an actual many-body ground state, e.g. in a superconductor. I believe this word can be easily replaced everywhere by “correlation”, which would resolve my confusion immediately (together with the mentioned schematics). I would like to insist to replace constructions like “triplet Cooper pair splitting” (title, summary, etc.) since this really suggests that the particles directly emerge from a superconducting phase with a triplet pairing. To me, having a source of triplet correlated electron pairs is at least as interesting and useful.

With these minor, but to me essential changes, I am happy to support the publication of this manuscript in Nature Communications.

Reviewer #3 (Remarks to the Author):

Thank you for clarifying the raised points. I find the revised manuscript suitable for publication in Nature Comms.

The reviews comments are in blue, our response in black and changes to the manuscripts are in red

We want to thank all three referee for their careful reading of our manuscript and the valuable comments. Please find below a point by point response.

REVIEWERS' COMMENTS

Reviewer #1 (Remarks to the Author):

The resubmitted paper by Wang, ten Haaf, Goswami and coworkers incorporates the comments by the reviewers in a way that makes the text more accessible. I feel that the revised manuscript has further improved. I am in favor of publication. Just one minor point: I would suggest to add a short note to the main manuscript that the information about how the values are extracted and the data processing methodology are detailed in supplementary Fig. S7.

We thank the referee for the time in reviewing our manuscript the second time and the positive assessment of the improvements. To address the minor point:

“The data extraction procedure is described in Fig. S7” is now added in the caption of Fig.4.

Reviewer #2 (Remarks to the Author):

The picture the authors seem to have – which is very plausible to me – is that the superconductor and the hybrid region is a near-perfect source of singlet electron pairs that then get transformed by spin orbit coupling (SOC) into triplets in the non-proximitized semiconductor region, and then detected in the Zeeman polarized QDs. I fully agree with this view, with the small question mark of why the effects of SOC should be stronger in the shorter semiconductor region than in the longer QD regions. But I am ok with this, since it is clearly described how this conclusion was reached. All minor points I made before were addressed, too.

This simple view could very easily be illustrated schematically in Fig. 1, with a singlet CPS pair, a subsequent spin rotation and the spin detection on the QDs. Maybe one can even put bra-kets to label the singlets and triplets at the corresponding positions.

My stumbling block was and still is that the wording in the abstract and in the introduction seems to suggest that there is a triplet pairing (these are the words), which the authors use in the sense of a triplet correlation or as triplet states between the QDs, but not in the sense of a superconducting triplet pairing. Most of my main concerns came from this use of the word “pairing” that in my view should be reserved for an actual many-body ground state, e.g. in a superconductor. I believe this word can be easily replaced everywhere by “correlation”, which would resolve my confusion immediately (together with the mentioned schematics). I would like to insist to replace constructions like “triplet Cooper pair splitting” (title, summary, etc.) since this really suggests that the particles directly emerge from a superconducting phase with a triplet pairing. To me, having a source of triplet correlated electron pairs is at least as interesting and useful.

With these minor, but to me essential changes, I am happy to support the publication of this manuscript in Nature Communications.

We thank the referee for the time in reviewing our manuscript the second time and the positive assessment of our work. The referee further clarifies his/her doubts about some phrases. Thus we made the following changes in the revised version:

Title: Triplet correlations in Cooper pair splitters realized in a two-dimensional electron gas

Abstract:

(third-last sentence)Not only do we observe a near perfect splitting of Cooper pairs into opposite-spin electrons (i.e. conventional singlet pairing), but also into equal-spin electrons, thus achieving triplet correlations between the quantum dots.

(second-last sentence)Importantly, the exceptionally large spin-orbit interaction in our 2DEGs results in a strong triplet component, comparable in amplitude to the singlet pairing.

(last sentence)The demonstration of ~~singlet and triplet~~ CPS in a scalable and flexible platform provides a credible route to study on-chip entanglement and topological superconductivity in the form of artificial Kitaev chains.

Main text:

Line 280: When $\mathbf{B} \perp \mathbf{B}_{\text{so}}$, C reaches close to 0 for both CAR and ECT, stressing that the triplet component can be tuned to be of similar magnitude to the conventional singlet pairing.

Line 287: Using spin-polarized quantum dots we performed spin-selective measurements of ECT and CAR and showed that the strong spin-orbit interaction in ternary 2DEGs results in comparable strengths of singlet and triplet correlations between the quantum dots.

Line 298: The demonstration of singlet and triplet correlations with Cooper pair splitters in 2DEGs paves the way for more advanced experiments to study entanglement and topological superconductivity.

Reviewer #3 (Remarks to the Author):

Thank you for clarifying the raised points. I find the revised manuscript suitable for publication in Nature Comms.

We thank the referee for the time in reviewing our manuscript the second time and the positive assessment of the improvements.